# ADAPTING LARGE LANGUAGE MODELS FOR CONTENT MODERATION: PITFALLS IN DATA ENGINEERING AND SUPERVISED FINE-TUNING

Warning: some content may contain racism, sexuality, or other harmful language.

## ABSTRACT

Nowadays, billions of people engage in communication and express their opinions on the internet daily. Unfortunately, not all of these expressions are friendly or compliant, making content moderation an indispensable task. With the successful development of Large Language Models (LLMs) in recent years, LLM-based methods have become a feasible solution for handling tasks in various domains. However, in the field of content moderation, there is still a lack of detailed work that systematically introduces implementation details. In this paper, we introduce how to fine-tune an LLM model that can be privately deployed for content moderation. Specifically, we discuss whether incorporating reasons during the fine-tuning process would be better or if it should be treated as a classification task directly. We also explore the benefits of utilizing reasons generated by more powerful LLMs for fine-tuning privately deployed models and the impact of different processing approaches when the answers generated by the more powerful LLMs are incorrect. We report the entire research process and the key findings in this paper, hoping to provide valuable experience for researchers who are fine-tuning privately deployed models in their domain-specific research.

## 1 INTRODUCTION

Content moderation is an indispensable topic in the realm of modern internet (Dinan et al., 2019; Jahan & Oussalah, 2023). Over the past decade, the field of internet content moderation has undergone a profound transformation, evolving from its early stages rooted in manual rule-based architectures (Kebriaei et al., 2023) to recent advancements characterized by deep learning-based fully automated moderation algorithms (Lilian Weng, 2023). The progress in deep learning technology has significantly enhanced the efficiency and precision of content moderation. Current automated content moderation architectures typically adhere to a discriminative paradigm (Deng et al., 2022; Husain & Uzuner, 2021). This paradigm involves the manual annotation of extensive, high-quality datasets, followed by supervised training of a discriminative model for content moderation. For a recent example, OpenAI details how they construct a holistic solution for real-world content moderation scenarios based on this paradigm, covering data preparation, model training, and deployment (Markov et al., 2023). While this paradigm has achieved significant success in various tasks such as text and multimodal moderation, and has helped businesses reduce substantial costs, it also exhibits several limitations: (1) Heavy Reliance on Data Annotation Quality (2) Limited Robustness to Out-of-Distribution Data in Open Environments (3) Lack of Interpretability.

These limitations become more pronounced when dealing with emerging scenarios like chatbots (Bhardwaj & Poria, 2023; Yu et al., 2023). For instance, in chatbot scenarios, there may be malicious users who attempt to manipulate the system by constructing inputs that bypass content moderation systems using techniques such as code-switching between multiple languages, using emojis, or employing variant characters. Traditional supervised discriminative approaches are clearly inadequate in addressing the dynamic and adversarial nature of these samples through data or optimization efforts alone (Manakul et al., 2023). On the other hand, recent successes of generative large language models (LLMs) in various complex reasoning and planning tasks naturally lead

Figure 1: Content Moderation with Auditing Processes. GPT-4 can provide complete auditing processes (left), but sometimes it presents limitations (right).

us to consider whether we can leverage the comprehensive grounding knowledge, strong language understanding, and logical reasoning capabilities inherent in LLMs to enhance the robustness and interpretability of content moderation tasks. The answer is clearly affirmative, and we have already witnessed some initial attempts and explorations. An interesting case in point is OpenAI's use of GPT-4 to optimize existing content moderation rating systems interactively (Lilian Weng, 2023), aiming to achieve optimal moderation performance. We have also made our own efforts by designing prompts to enable GPT-4 to perform automatic content moderation on real-world data (As Fig. 1 shows). It is evident that GPT-4 not only produces reasonable results but can also provide a complete auditing process. However, for most companies and organizations, reliance on third-party GPT-4 models still presents significant limitations: (1) Compliance Requirements: There are regulatory and compliance requirements that may constrain the use of third-party models for content moderation, (2) Cost Considerations, and (3) Domain-Specific Knowledge Injection. Addressing these limitations and developing custom content moderation solutions that meet specific compliance, cost, and domain requirements remains an ongoing area of research and development in the field.

One viable technical approach involves constructing labeled samples with an associated moderation process (moderation reasoning chain) and then performing supervised fine-tuning (SFT) on open-source large language models to align them with the specific requirements of a given content moderation task. Over the past year, this approach has demonstrated success in various tasks beyond content moderation, such as solving mathematical problems and logical reasoning (Gao et al., 2023; Luo et al., 2023a;b). Researchers have proposed a variety of techniques to improve the ultimate performance of this approach, including but not limited to data engineering, audit chain reliability assessment, optimization objectives, and more (Gerrard, 2018; Morrow et al., 2022; Singhal et al., 2023). The effectiveness of these techniques in real-world content moderation scenarios still requires validation. A relevant work in this context comes from AWS (James et al., 2023), where they attempt to fine-tune large models directly using label information (without the audit reasoning chain). However, in our practical experience, we find that this approach exhibits poor out-of-distribution robustness and lacks interpretable moderation process outputs.

By adapting LLMs to the content moderation, the ideal case is to build a model that is significantly more robust than previous discriminative models simply by applying supervised fine-tuning (SFT) to LLMs (Hajibabaee et al., 2022; Rizwan et al., 2020). However, the practical implementation of such a straightforward idea is fraught with numerous design challenges. Various design choices, including data engineering and training methodologies, have a profound impact on the final model's performance. In this paper, we aim to take the first step by conducting large-scale experiments to assess the feasibility of applying large language models to content moderation scenarios and meticulously analyze the effectiveness of various commonly used techniques throughout the entire pipeline. Specifically, at the data engineering level, we adapt a weak supervision paradigm to reduce the cost of manual annotation. This involves expert annotation of the final labels for samples, with

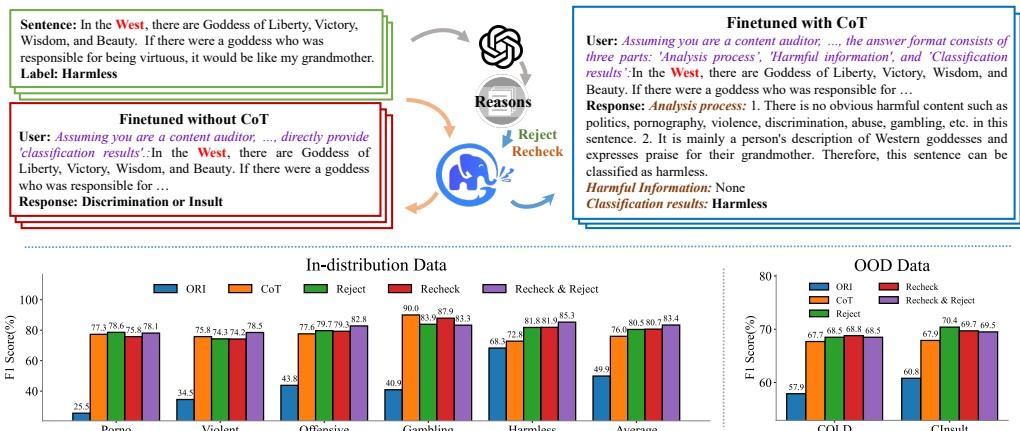

Figure 2: Framework of fine-tuing privately deployed content moderation model. Fine-tuning LLMs with the pattern $(x_i, y_i)$ can lead to severe overfitting. For instance, sentences that contain regional characteristics will be classified into Discrimination or Insult. Incorporating an auditing process into the fine-tuning stage can effectively overcome this issue.

GPT-3.5 & GPT-4 handling the more labor-intensive moderation reasoning chain. In contrast to many existing works that adapt fully unsupervised paradigms (where all data is generated by GPT-3.5 & GPT-4), our approach allows us to leverage human-generated golden labels for reflection on the reasoning chain (see the methods section for details). Additionally, we incorporate conventional optimization strategies such as data deduplication (He et al., 2010). During the training process, we observe common hallucination issues in GPT models when generating the moderation reasoning chain for production. These hallucinations can lead to erroneous labels. Here, we leverage weak supervision information to correct the entire process. Applying this entire pipeline to real-world data, we have arrived at several key findings: (1) Incorporating reasoning processes during the fine-tuning will enhance the robustness of the model and effectively overcome overfitting. In addition to the labels, reasoning process is necessary for fine-tuning. (2) Introducing weak supervision can effectively filter out samples with poor quality in reasoning processes, improve the quality of the fine-tuning data, and enhance the performance of the fine-tuned model. (3) The capabilities of the models limit their fine-tuned performance. For example, replacing GPT-3.5's auditing processes with GPT-4's more detailed auditing processes, the improvement in performance is slight for models with weaker capabilities compared to models with stronger capabilities.

## 2 CONTENT MODERATION SYSTEM BASED ON LLMS

### 2.1 NOTATIONS

Given the raw real-world dataset $\mathcal{D}_{raw} = \{x_i, y_i\}_{i=1}^N$, where $x_i$ is the input sentence of the $i$-th sample, and $y_i \in \{1, \cdots, K\}$ is the corresponding *weak supervision* label, which only consists of the labels annotated by humans, without including the complete reasoning processes. Then we augment the raw training dataset by incorporating a reasoning process, yielding $\mathcal{D} = \{x_i, r_i, y_i\}_{i=1}^N$, where $r_i$ is the reason behind the classification of $x_i$ as $y_i$. However, obtaining the exact $r_i$ is inherently challenging, so we approximate it through LLMs. Specifically, given an input $x_i$, we prompt cloud LLM services to generate the analysis process response $\hat{r}_i$ and corresponding predicted label $\hat{y}_i$, and we denote the augmented training dataset as $\mathcal{D}_{train} = \{x_i, \hat{r}_i, \hat{y}_i\}_{i=1}^N$.

### 2.2 DATA DEDUPLICATION

Previous research has demonstrated that repetitive data lacking diversity does not contribute to the improvement of training models. What models truly require are carefully selected data samples that exhibit diversity (Liang et al., 2023). Eliminating highly similar data not only reduces the time required for model training but also minimizes the inefficiency caused by redundant reasoning

processes. Consequently, we conduct deduplication on the collected real-world dataset to eliminate repetitive data that does not benefit the training of the model. Specifically, we utilize a pretrained text encoder to obtain embeddings for each sentence (Liu et al., 2021). Subsequently, we perform clustering on the data distribution within each category and select one sample from each cluster, aiming to obtain samples with distinct semantic characteristics. Following this approach, we curate a golden set consisting of 145 examples. To ensure the quality of this selected data, we conduct manual label recheck on the golden set.

### 2.3 FINE-TUNING LLMs WITHOUT CoT

Directly employing the cloud LLM service API (OpenAI, 2023) for content moderation is not feasible, particularly when handling a substantial volume of private data in numerous content moderation scenarios. In such cases, utilizing a privately deployed LLM for content moderation becomes essential to accomplish the task. However, the present performance of open-source pre-trained models (Du et al., 2022; Zeng et al., 2022) is inadequate in domain-specific tasks, such as content moderation. This necessitates alignment fine-tuning for vertical domain tasks.

One of the simplest approaches is to perform SFT on open-source LLMs with the raw training set $\mathcal{D}_{raw} = \{x_i, y_i\}_{i=1}^{N}$, and the model is only required to predict $y_i$ given $x_i$, and we get the private classifier $\mathcal{M}_p$.

However, a major drawback of this method is its high susceptibility to catastrophic overfitting on limited data and large-scale networks, and the experiments results are shown in Sec. 3.2.2. To address this issue, we introduce fine-tuning with CoT inspired by previous research in mathematical reasoning (Luo et al., 2023a; Gao et al., 2023).

### 2.4 FINE-TUNING LLM WITH CoT

In this section, we present the fine-tuning of LLM with CoT to incorporate reasoning processes during training. However, acquiring the reasoning $r$ is challenging, while obtaining input $x$ and label $y$ in real-world data is comparatively easier. Conventional methods for procuring reasoning involve manual annotation, which introduces two issues: (1) the high cost and slow pace of human annotation, making it difficult to update training data continuously, and (2) inconsistencies in annotation standards and reasoning logic among different individuals, making it challenging to establish a standardized and consistent reasoning process.

The emergence of cloud LLM services (denoted as $\mathcal{M}_s$) addresses the above issues. First, although the cloud LLM services provided by large model service providers may not be directly deployed in actual businesses, they can still generate some of reasoning data with a few API calls. Second, the judgment criteria of LLMs are unified, ensuring consistency in reasoning processes across different samples and maintaining the standardization of these processes. We can take advantage of cloud LLM services to generate reasoning processes for augmenting our training set. Subsequently, using the training set with reasons, we can fine-tune the open-source LLMs for specific domain tasks, thereby enhancing the capabilities of privately deployed models ($\mathcal{M}_p$) for practical real-world tasks.

**Weak supervision.** To obtain the reasoning process why a training sample $x_i$ is classified as class $y_i$, a natural approach is to prompt the $\mathcal{M}_s$ with sentence $x_i$ and its corresponding label $y$, and then require the $\mathcal{M}_s$ to analyse the reasoning process $\hat{r}_i$. However, previous studies have highlighted the issue of hallucinations in LLMs (Manakul et al., 2023; Bang et al., 2023). If $\mathcal{M}_s$ is given access to the true label before analyse, its response may be influenced by hallucinations. This would significantly compromise the quality of the data for fine-tuning the model. Consequently, we propose a weak supervision approach in which $\mathcal{M}_s$ is not provided with the true label $y_i$ during its reasoning process. Instead, it generates predicted labels $\hat{y}_i$ based on its reasoning process, and the true labels from the training set are only used to assess whether the judgments of $\mathcal{M}_s$ are correct.

Specifically, we can achieve this through the following steps:

1. Determine the instruction that includes the reasoning process based on the actual task and training dataset $\mathcal{D}_{raw} = \{x_i, y_i\}_{i=1}^{N}$.

2. Prompt the instruction and training data $x_i$ to a remote model service $\mathcal{M}_s$ to obtain the reasoning process $\hat{r}_i$ and predicted label $\hat{y}_i$.

3. Fine-tune the open-source model using the training set $\mathcal{D}_{train} = \{x_i, \hat{r}_i, \hat{y}_i\}_{i=1}^{N}$ that includes reasoning to get privately deployed model $\mathcal{M}_p$.

However, although the model capabilities offered by LLM service providers surpass those of open-source models, these cloud model services might incorrectly classify the training data, which makes it difficult to acquire accurate reasons for training privately deployed models. To address this issue, we propose various strategies to clean the dataset when an incorrect response is obtained. Specifically, we select samples in which the $\mathcal{M}_s$ predicts inaccurately by examining the consistency between the predicted results and the ground truth. For samples with prediction errors, that is, samples where $\hat{y} \neq y$, we have two distinct strategies for managing them.

(a) Removing samples that are misclassified by $\mathcal{M}_s$, which is a simple and natural solution. The specific process involves inserting the following step between the step 2 and step 3:

2.a. If $\hat{y}_i \neq y_i$, remove the sample $< x_i, \hat{r}_i, \hat{y}_i >$ from the training dataset $\mathcal{D}_{train}$.

(b) Introducing self-recheck to rectify the model's outputs. The specific process entails incorporating the subsequent step between steps 2 and 3:

2.b. If $\hat{y}_i \neq y_i$, prompt $\mathcal{M}_s$ to recheck the response with the original question and the incorrect reason $\hat{r}_i$ and predicted label $\hat{y}_i$, then update the revised reason $\hat{r}_i$ and the prediction $\hat{y}_i$.

Though the self-recheck process can be repeated multiple times to obtain the correct prediction result, we perform the reflection once during the experiment. If $\mathcal{M}_s$ persists in making errors even after re-answering, it indicates the question is too hard for $\mathcal{M}_s$, and compelling it to change the response might result in lower-quality answers due to hallucinations.

## 3 EXPERIMENTS

### 3.1 EXPERIMENTS SETTING

**Dataset:** The business scenario in the experiment focuses on content moderation in a Chinese environment. We classify the content into 5 categories: Pornography (Porno), Violence, Discrimination or Insult (Offensive), Gambling, and Harmless. The data originates from our real-world business scenarios and has been annotated with labels $y$ by human annotators. To validate the models' business performance and generalization ability, we evaluate their performance on both in-distribution data and out-of-distribution (OOD) data. Since the primary business scenario in this experiment involves the Chinese context, we utilize the open benchmark dataset COLD (Deng et al., 2022) and the Chinese insult dataset CInsult[1] as the OOD data. (Details please refer to Appendix E). **Model:** For open-source base models, we choose ChatGLM2-6B (Du et al., 2022; Zeng et al., 2022) and Baichuan-13B-Chat as the base models, because they have been reported good performance in basic Chinese task and demonstrate a more prominent understanding of the Chinese language context compared to LLaMA (Touvron et al., 2023). Although some researches align LLaMA for Chinese [2] [3], they lack official formal evaluations and guarantees, making them unsuitable as base models. **Fine-tuning:** Fine-tuning the model can be done using methods including full-parameter-tuning, P-tuning (Liu et al., 2022), or LORA (Hu et al., 2021). Parameter-Efficient Fine-Tuning (PEFT) has been proven effective in preventing overfitting in models, and several studies have demonstrated that PEFT outperforms full-parameter-tuning (Zeng et al., 2023; Mangrulkar et al., 2022). In the officially released code for fine-tuning ChatGLM2[4], parameter-effective P-tuning-V2 is recommended. On the other hand, tuning the Baichuan-Chat based on LORA is among the widely recommended PEFT methods. Therefore, we chose P-tuning-V2 for fine-tuning the ChatGLM and fine-tuning the Baichuan-Chat based on LORA.

---

[1]https://github.com/wjx-git/IllegalTextDetection

[2]https://github.com/ymcui/Chinese-LLaMA-Alpaca

[3]https://huggingface.co/shibing624/chinese-llama-plus-13b-hf

[4]https://github.com/THUDM/ChatGLM2-6B/tree/main/ptuning

Table 1: Results on the different models fine-tuned with different strategies, the best-performing F1 Score for each harm type has been highlighted in bold for each model. "Average" indicates the micro-average performance on in-distribution.

| Model | Strategies | Metrics | In-distribution | | | | | | OOD | |
|---|---|---|---|---|---|---|---|---|---|---|
| | | | Porno | Violent | Offen. | Gamb. | Harml. | Average | COLD | CInsult |
| **GPT-3.5** | - | Recall | 80.0 | 82.7 | 92.0 | 97.3 | 59.9 | 73.0 | 63.5 | 78.3 |
| | | Precision | 95.2 | 80.5 | 83.6 | 96.1 | 91.8 | 89.3 | 73.0 | 83.5 |
| | | F1 Score | 87.0 | 81.6 | 87.6 | 96.7 | 72.5 | 80.3 | 60.3 | 79.0 |
| **ChatGLM2 (6B)** | **Original** | Recall | 44.0 | 64.0 | 57.3 | 76.0 | 63.6 | 61.9 | 57.5 | 54.1 |
| | | Precision | 17.9 | 23.6 | 35.4 | 27.9 | 73.6 | 41.7 | 64.0 | 70.9 |
| | | F1 Score | 25.5 | 34.5 | 43.8 | 40.9 | 68.3 | 49.9 | 57.9 | 60.8 |
| | **Setting A** | Recall | 68.0 | 81.3 | 74.0 | 84.0 | 63.4 | 69.2 | 65.3 | 61.5 |
| | | Precision | 89.5 | 70.9 | 81.6 | 96.9 | 85.4 | 84.2 | 73.7 | 76.4 |
| | | F1 Score | 77.3 | 75.8 | 77.6 | **90.0** | 72.8 | 76.0 | 67.7 | 67.9 |
| | **Setting B** | Recall | 73.3 | 82.7 | 80.0 | 80.0 | 76.7 | 77.8 | 67.7 | 66.4 |
| | | Precision | 84.6 | 67.4 | 79.5 | 88.2 | 87.6 | 83.4 | 72.4 | 76.4 |
| | | F1 Score | **78.6** | 74.3 | 79.7 | 83.9 | 81.8 | 80.5 | 68.5 | **70.4** |
| | **Setting C** | Recall | 66.7 | 78.7 | 79.3 | 82.7 | 78.7 | 78.1 | 68.0 | 64.5 |
| | | Precision | 87.7 | 70.2 | 79.3 | 93.9 | 85.3 | 83.4 | 73.3 | 76.9 |
| | | F1 Score | 75.8 | 74.2 | 79.3 | 87.9 | 81.9 | 80.7 | **68.8** | 69.7 |
| | **Setting D** | Recall | 66.7 | 82.7 | 80.0 | 73.3 | 84.7 | 81.0 | 67.6 | 65.6 |
| | | Precision | 94.3 | 74.7 | 85.7 | 96.5 | 85.8 | 86.0 | 72.7 | 75.9 |
| | | F1 Score | 78.1 | **78.5** | **82.8** | 83.3 | **85.3** | **83.4** | 68.5 | 69.5 |
| **Baichuan (13B)** | **Original** | Recall | 74.7 | 44.0 | 48.0 | 54.7 | 59.2 | 56.8 | 41.6 | 47.5 |
| | | Precision | 47.5 | 35.9 | 55.4 | 48.2 | 79.0 | 61.5 | 70.0 | 77.4 |
| | | F1 Score | 58.0 | 39.5 | 51.4 | 51.2 | 67.7 | 59.0 | 51.9 | 58.9 |
| | **Setting A** | Recall | 70.7 | 86.7 | 85.3 | 98.7 | 45.2 | 63.4 | 65.7 | 76.2 |
| | | Precision | 86.9 | 63.7 | 81.5 | 84.1 | 85.7 | 81.1 | 72.1 | 83.7 |
| | | F1 Score | 77.9 | 73.4 | **83.4** | 90.8 | 59.2 | 71.2 | 66.3 | **79.8** |
| | **Setting B** | Recall | 78.7 | 85.3 | 80.7 | 96.0 | 82.9 | 83.5 | 66.3 | 72.5 |
| | | Precision | 89.4 | 63.4 | 85.8 | 92.3 | 90.3 | 86.2 | 73.8 | 83.0 |
| | | F1 Score | **83.7** | 72.7 | 83.2 | **94.1** | **86.5** | **84.9** | **69.0** | 77.4 |
| | **Setting C** | Recall | 69.3 | 89.3 | 79.3 | 98.7 | 76.1 | 79.3 | 69.0 | 77.2 |
| | | Precision | 77.6 | 64.4 | 77.3 | 82.2 | 87.7 | 81.3 | 71.3 | 82.0 |
| | | F1 Score | 73.2 | **74.9** | 78.3 | 89.7 | 81.5 | 80.3 | **69.0** | 79.5 |
| | **Setting D** | Recall | 77.3 | 86.7 | 80.7 | 88.0 | 80.0 | 81.2 | 68.1 | 73.6 |
| | | Precision | 81.7 | 60.2 | 80.7 | 80.5 | 88.0 | 81.7 | 72.2 | 79.8 |
| | | F1 Score | 79.5 | 71.0 | 80.7 | 84.1 | 83.9 | 81.5 | **69.0** | 76.4 |

### 3.1.1 DIFFERENT TRAINING STRATEGIES

We apply various fine-tuning strategies, such as reject strategy, recheck strategy, and a combination of both strategies, and the prompt templates please refer to Appendix A.

**GPT-3.5:** Ideally, we hope the privately deployed model to achieve competitive performance comparable to the cloud LLM service. To evaluate this, we test the performance of GPT-3.5-turbo using the same instructions, which involves conducting content analysis before making a judgment. ***Please note that** GPT-3.5 has a content management policy. Some content is filtered by content filtering policies, and in such cases, we consider it as a successful detection by GPT-3.5 if the responses of positive samples are filtered.*

**Original:** We evaluate the ChatGLM2-6B and Baichuan-13B-Chat models without any fine-tuning for content moderation as the baseline, to offer an intuitive comparison of the fine-tuning effects achieved through various strategies.

**Setting A:** We generate CoT $\hat{r}$ and predicted class $\hat{y}$ using GPT-3.5 on the training set. Then, we utilize $\mathcal{D}_{train} = \{x_i, \hat{r}_i, \hat{y}_i\}_{i=1}^{N}$ to fine-tune ChatGLM2-6B and Baichuan-13B-Chat.

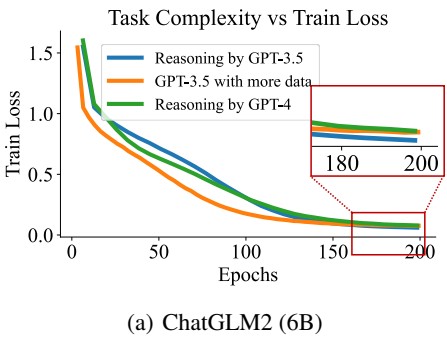 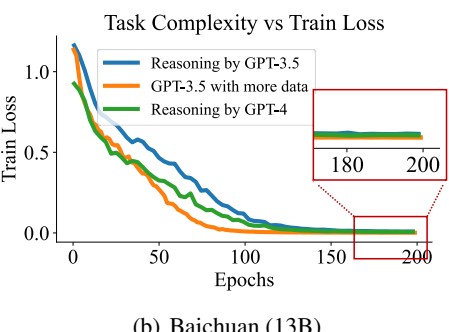

(a) ChatGLM2 (6B)     (b) Baichuan (13B)

Figure 3: Task Complexity for models with different capability. "Reasoning by GPT-3.5" indicates the reasoning processes ($r$) in train dataset is generated form GPT-3.5, and "GPT-3.5 with more data" indicates the model is fine-tuned with LLM-expanded training samples ($x$).

**Setting B:** Taking into account that the answers generated by GPT-3.5 may be incorrect, we check GPT-3.5's responses by the ground truth. We enhance the quality of the training data by discarding samples in which GPT-3.5 made inaccurate predictions (i.e., $\hat{y} \neq y$), as demonstrated in step 2.a.

**Setting C:** In addition to discarding samples with incorrect predictions, we instruct GPT-3.5 to perform self-rechecks and modify its responses for these samples as shown in step 2.b.

**Setting D:** This strategy combines the recheck and reject approaches. Firstly, the model reflects on and modifies incorrect answers. Subsequently, samples that persist as incorrect after re-answering are removed from the training dataset.

## 3.2 RESULTS

Table 2: F1 Score of models fine-tuned with GPT-3.5 & GPT-4-generated reasoning processes.

| Base Models | $\mathcal{M}_s$ | Porno | Violent | Offensive | Gambling | Harmless | Average |
|---|---|---|---|---|---|---|---|
| ChatGLM2 (6B) | GPT-3.5 | 77.3 | 75.8 | 77.6 | 90.0 | 72.8 | 76.0 |
| | GPT-4 | 81.8 | 68.3 | 72.8 | 79.7 | 81.9 | 79.1 |
| Baichuan (13B) | GPT-3.5 | 77.9 | 73.4 | 83.4 | 90.8 | 59.2 | 71.2 |
| | GPT-4 | 70.9 | 74.5 | 79.0 | 93.5 | 87.3 | 83.9 |

We evaluate the performance of all models, including the original model, and the results are displayed in Table 1. We also highlight the best-performing models in terms of the F1 Score metric of each model (F1 Score is a good indicator of overall performance, we only emphasize this metric to make the table more readable). From the Table 1, it can be observed that although these models perform poorly before fine-tuning, all fine-tuned models are able to achieve performance comparable to GPT-3.5. Particularly in the category of harmless, all fine-tuned models significantly surpass GPT-3.5's performance.

### 3.2.1 DESIGN CHOICE OF DATA ENGINEERING

**Expanding training sentences ($x$) by LLM does not benefit model fine-tuning.** Previous works generate $x$ for fine-tuning by LLMs to expand the training set (Luo et al., 2023a), but in the context of content moderation, the $x$ generated form LLMs does not significantly help with fine-tuning the model. In our experiments, we initially selected 145 high-quality data samples as golden set. Then, we expand it with over 1000 new training samples generated by GPT-3.5. From the generated data, we select an additional 290 samples with the highest diversity (using sentence embedding clustering, details please refer to Appendix B) to expand the training data. As shown in Table 3, we find that augmenting the training set results in several times more training costs, but it does not

provide significant benefits to fine-tuning the model. For ChatGLM2-6B, expanding samples make it more difficult for the model to decrease its loss to the same level as on the Golden set (Fig. 3), and expended training data doesn't improve the model's performance. As for Baichuan-13B, although the model's training loss can decrease to the same level as on the Golden set, expanded examples improve the performance very slightly.

Table 3: Fine-tuning without & with LLM-generated training data.

| Models | Porno | Violent | Offensive | Gambling | Harmless | Average |
|---|---|---|---|---|---|---|
| ChatGLM2 | 77.3 vs 82.4 | 75.8 vs 77.4 | 77.6 vs 77.8 | 90.0 vs 93.2 | 72.8 vs 65.3 | 76.0 vs 73.3 |
| Baichuan | 77.9 vs 81.4 | 73.4 vs 77.3 | 83.4 vs 75.2 | 90.8 vs 91.2 | 59.2 vs 65.3 | 71.2 vs 72.8 |

**Obtain reasoning processes ($\hat{r}$) form different cloud LLM services.** We utilize both GPT-4 and GPT-3.5 to generate reasoning processes on the same training dataset for fine-tuning. As expected (Table 4), the auditing processes generated by GPT-4 were more detailed (Length) and accurate (Macro F1). As shown in Fig. 3, it is difficult for the model with weak capability (ChatGLM2-6B) to fit the detailed reasoning processes. We find that the training loss of the model could not decrease to the level achieved by fine-tuning with the reasoning generated by GPT-3.5.

Table 4: The comparison of training data augmented by GPT-3.5 and GPT-4.

| Reason from | Length | Macro F1 |
|---|---|---|
| GPT-3.5 | 188 | 72.9 |
| GPT-4 | 210 | 79.3 |

Consequently, the performance improvement on the test dataset of ChatGLM2 is also slight. On the other hand, from Fig. 3, we can find the training loss of Baichuan-13B quickly decreases to the same level comparable to the fine-tuning loss achieved with the reasoning generated by GPT-3.5, and the larger model, Baichuan-13B, shows a significant improvement when fine-tuned with the reasoning processes generated by GPT-4.

### 3.2.2 DESIGN CHOICE OF SFT

**Fine-tuning LLMs with the pattern $(x_i, y_i)$ can lead to severe overfitting.** As shown in Fig. 4, after fine-tuning without incorporating reasons, the model's performance significantly deteriorates on out-of-distribution (OOD) data. Specifically, Fig. 4 displays the recall rates on negative samples of different models on binary classification OOD benchmark data. "COLD-ChatGLM" indicates the recall on the COLD dataset (Deng et al., 2022) using the ChatGLM model, while "ORI" and "w/o CoT" denote the

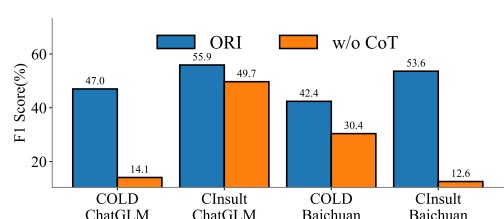

Figure 4: Recall on OOD datasets.

original model without fine-tuning and the model fine-tuned without incorporating reasons, respectively. In other words, the model fine-tuned without incorporating reasons exhibits severe overfitting (details please refer to Appendix D), making it unsuitable for deployment in practical business scenarios.

**Weak supervision can improve the fine-tuning performance.** Compared to the unsupervised setting (Setting A), we introduce weak supervision methods to improve the quality of reasoning processes in the training set. Specifically, we find that (1) filtering low-quality reasoning processes through weak supervision can help enhance fine-tuning performance. As shown in Table 1 (Setting B), the model's performance significantly improves after filtering out low-quality samples. (2) Rechecking low-quality reasoning processes through weak supervision can also enhance the quality of the fine-tuning data. As shown in Table 1 (Setting C), the model's performance improves by regenerating reasoning processes for low-quality samples. Moreover, these two strategies are not conflicting. When both strategies are combined (Setting D), the model achieves similar or even better results than the best performance of either strategy alone (The results of models fine-tuned with reasoning processes generated form GPT-4 please refer to Appendix F).

## 4 RELATED WORK

**Content Moderation.** Content moderation plays a vital role in ensuring the security of online content, garnering widespread attention in recent years. This attention encompasses both academic and industry product research.

For researches on content moderation, numerous typical tasks have been proposed, accompanied by impressive performance reports. • **Discrimination-based methods:** OpenAI introduces a discrimination-based method employing active learning and adversarial training strategies (Markov et al., 2023). They emphasize the importance of meticulous data selection, the necessity of active learning, and the multitude of challenges arising from data imbalance. • **LLM-based Methods** (Li et al., 2023a; Franco et al., 2023; Mullick et al., 2023): By utilizing the ChatGPT API, assuming ChatGPT as a content moderator, conducting content auditing in a conversational format. There are also many specific methods in the practical deployment of content moderation in the industry. • **Expert Rules** (Kebriaei et al., 2023): Aggressive language filtering in the maintenance of social platform security often relies on keyword matching, leveraging prior expert knowledge to pre-define expert rules and facilitate content moderation. • **Policy Refinement** (Lilian Weng, 2023): Policy experts draft content policy and assign labels to a small number of examples based on the policy to create a golden dataset. Subsequently, GPT-4 is instructed to read the content policy, assign labels to the same dataset, and scrutinize the discrepancies between GPT-4's judgments and human judgments. This process involves analyzing the ambiguity in policy definitions and providing further policy clarification accordingly, and iterations of this process continue until the desired policy quality is achieved.

**Fine-tuning LLM with Chain of Thought.** The Chain of Thought (CoT) has been introduced as a viable method for achieving interpretability in language models (Wei et al., 2022; Wang et al., 2022). A crucial criterion for interpretability methods is their ability to accurately capture the cognitive processes occurring within the model.

CoT effectively enhances both the accuracy and interpretability of the model by employing logical analysis and reasoning (Suzgun et al., 2022). Recent studies integrate inference processes into the fine-tuning models, aiming to move beyond solely fitting the final answer, especially in math reasoning. • **Process Supervision**: Uesato et al. (Uesato et al., 2022) conduct the initial comprehensive comparison between process-based and outcome-based supervisions implemented on a natural language task. A recent study conducted by OpenAI (Lightman et al., 2023) examines the distinction between outcome-supervised reward models (ORMs) and process-supervised reward models (PRMs). ORMs rely solely on the final result of the model's response, whereas PRMs receive feedback at each step of the chain-of-thought. The findings demonstrate that process supervision is capable of training significantly more dependable reward models compared to outcome supervision. • **Fine-tuning with CoT**: Yuan et al. (Yuan et al., 2023) generate CoT-augmented samples without any trained process-level reward models and outperform the supervised fine-tuning (SFT) accuracy. Hsieh et al. (2023) introduce Distilling step-by-step to train smaller models with less training data, which reduces the amount of training data required for fine-tuning and distillation of LLMs into smaller model sizes.

Although attempts have been made to fine-tune large models in many fields, including financial (Yu, 2023), legal (Huang et al., 2023), medical (Xiong et al., 2023), educational (Dan et al., 2023), astronomical (Li et al., 2023b), there hasn't been a specific work that provides a detailed explanation for content moderation.

## 5 CONCLUSION

In this paper, we conduct a fine-tuning attempt to privately deploy LLM with content moderation as a specific scenario. We enhance the model's generalization ability by introducing reasoning process during the fine-tuning process, resulting in the privately deployed model achieving performance that is not weaker than that of GPT-3.5 on content moderation. Additionally, we introduce weak supervision to improve the reasoning quality of the training dataset, which improve the fine-tuning performance at a small annotation cost. We also discuss the different design choice of data engineering and SFT. Overall, we believe the research offers valuable insights for researchers in various fields to fine-tune privately deployed models in their domains.

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

# A    TEMPLATES IN THIS PAPER

Table 5: Templates of different strategies.

| Strategy | Template |
|---|---|
| Classification | QUERY: [Classification Instruction] $x_i$
RESPONSE: $\hat{y}_i$ |
| With COT | QUERY: [Classification with COT Instruction] $x_i$
RESPONSE: Reason: $\hat{r}_i$
      Classification: $\hat{y}_i$ |
| COT with Self-check | QUERY: [Classification with COT Instruction] $x_i$
RESPONSE: Reason: $\hat{r}_i$
      Classification: $\hat{y}_i$
If $\hat{y}_i \neq y_i$:
    RE-QUERY: [Self-check Instruction]
    RESPONSE: Reason: $\hat{r}_i$
         Classification: $\hat{y}_i$ |

### [Classification Instruction]

```
Assuming you are a professional content auditor, determine the harmfulness of the following
content and classify it into one or more of the alternative categories: [Political
Harmful, Pornography, Violence, Discrimination or Insult, Gambling, Harmless], and provide
a "classification result" directly. The answer format is: 'Classification result:'
The sentence to be analyzed is: #sentence xᵢ#
```
**Response:** Classification results: #prediction $\hat{y}$#

### [Classification with COT Instruction]

```
Assuming you are a professional content auditor, determine the harmfulness of the following
content and categorize it into one or more categories within [Political Harmful, Pornography,
Violence, Discrimination or Insult, Gambling, Harmless]. First, explain in detail the reasons
for the classification of the content in the "analysis process". The analysis process should
be derived in the form of a thought chain, and the derivation process must be detailed. Then
extract 'harmful information' and extract harmful content from the sentence, and then provide
'classification results'. The answer format consists of three parts: 'Analysis process','
Harmful information ', and' Classification results'.
The sentence to be analyzed is: #sentence xᵢ#
```
**Response:** Analysis process: <Analysis process>
Harmful information: <Harmful information>
Classification results: #prediction $\hat{y}$#

### [Self-check Instruction]

```
The 'classification result' in the analysis is incorrect. Please review the paragraph #
sentence xᵢ# in the required format to correct the answer.
```
**Response:** Analysis process: <Analysis process>
Harmful information: <Harmful information>
Classification results: # prediction $\hat{y}$ #

# B    GENERATE NEW TRAIN DATA BY CHATGPT.

We have expanded the golden set to increase the size of the training dataset. Specifically, we used role-playing instructions to prompt ChatGPT to generate descriptive content for each category in the training set. Although ChatGPT may occasionally refuse to answer, we still obtained thousands of data points. Next, we encoded the sentences from each category to obtain embeddings, and then performed clustering for each category separately to extract the sentences with the highest diversity. Finally, we applied the same processing using GPT-3.5 and GPT-4 to these selected sentences to obtain reasoning processes. Through this approach, we were able to triple the size of the original training set.

And the template is:

```
#masked instruction#, please generate ten more statements containing #harm type#, which will
be used legally.
Response:  #new harmful sentences#
Note:  we mask the details of the instruction for this task, as it would lead LLMs to generate
harmful sentences, which could potentially have negative impacts.
```

## C  MODIFYING COT BASED ON THE ORIGINAL MODEL'S RESPONSE

Previous fine-tuning attempts have shown that reducing changes to the original distribution of model outputs during fine-tuning can lead to better performance. Therefore, we tried correcting incorrect answers by GPT-3.5 and GPT-4 based on the original model outputs. Specifically, we first had the base model generate replies with reasoning processes on the training set. Then we selected the incorrect parts from these replies. We inputted these samples along with the base model's replies into GPT-3.5 and GPT-4, allowing them to correct the incorrect answers and modify the reasoning process. Afterward, we updated the modified answers in the training set to fine-tune the base model.

And the template is:

```
Assuming you are a professional content auditor, determine the harmfulness of the following
content and classify it into one or more of the alternative categories: [Political
Harmful, Pornography, Violence, Discrimination or Insult, Gambling, Harmless], and provide
a "classification result" directly.  The answer format is:  'Classification result:'
The sentence to be analyzed is:  #sentence $x_i$#
An assistant's answer is:  #response of base model#
The answer is incorrect.  Please modify the assistant's answer according to the **formatting
requirements** to ensure that the sentences are correctly classified.
Response:  Analysis process:  <Analysis process>
Harmful information:  <Harmful information>
Classification results:  #prediction $\hat{y}$#
```

## D  FINE-TUNING THE MODEL WITHOUT COT

One approach to fine-tune the open-source models to align with specific vertical domain tasks is to perform SFT on open-source LLMs using the raw training set in the partten of $(x_i, y_i)$. However, this method has a significant drawback, as it tends to overfit when data is limited and networks are large-scale.

Table 6: Results on the different models fine-tuned without reasoning processes. "Average" indicates the micro-average performance on in-distribution and F1 Score below 50 indicates that the model's performance is worse than random guessing.

| Model | Metric | In-distribution | | | | | | OOD | |
|---|---|---|---|---|---|---|---|---|---|
| | | Porno | Violent | Offen. | Gamb. | Harml. | Average | COLD | CInsult |
| **ChatGLM2-6B** | Recall | 90.7 | 92.0 | 86.0 | 92.0 | 94.2 | 92.0 | 50.8 | 42.7 |
| | Precision | 88.3 | 87.3 | 93.5 | 97.2 | 96.4 | 94.3 | 70.3 | 77.6 |
| | F1 Score | 89.5 | 89.6 | 89.6 | 94.5 | 95.3 | 93.1 | 42.6 | 55.0 |
| **Baichuan-13B** | Recall | 92.0 | 94.7 | 96.0 | 94.7 | 94.2 | 94.4 | 56.5 | 42.7 |
| | Precision | 95.8 | 94.7 | 90.0 | 98.6 | 98.4 | 96.2 | 72.8 | 74.5 |
| | F1 Score | 93.9 | 94.7 | 92.9 | 96.6 | 96.3 | 95.3 | 55.5 | 41.3 |

# E  DETAILS OF OOD DATASETS

we utilize the open benchmark dataset COLD (Deng et al., 2022) and the Chinese insult dataset CInsult[5] as the OOD data.

COLD contains 37,480 comments with binary offensive labels and covers diverse topics of race, gender, and region. To gain further insights into the data types and characteristics, we annotate the test set at a fine-grained level with four categories: attacking individuals, attacking groups, anti-bias and other non-offensive. We take the test set (5323 samples) as OOD dataset for evaluation.

CInsult is a dataset contains 7137 abuse comments, and there are two versions of the dataset, original version and augmented version, the augmented version has been processed to separate the words, making it more challenging. In the experiment, we chose a more challenging version.

# F  OTHER RESULTS

Table 7: Results on the different models fine-tuned with different strategies (Baichuan-13B, reasoning processes generated from GPT-4).

| Strategies | Metrics | In-distribution | | | | | | OOD | |
|---|---|---|---|---|---|---|---|---|---|
| | | Porno | Violent | Offen. | Gamb. | Harml. | Average | COLD | CInsult |
| **Original** | Recall | 74.7 | 44.0 | 48.0 | 54.7 | 59.2 | 56.8 | 41.6 | 47.5 |
| | Precision | 47.5 | 35.9 | 55.4 | 48.2 | 79.0 | 61.5 | 70.0 | 77.4 |
| | F1 Score | 58.0 | 39.5 | 51.4 | 51.2 | 67.7 | 59.0 | 51.9 | 58.9 |
| **Setting A** | Recall | 60.0 | 76.0 | 75.3 | 96.0 | 85.4 | 81.4 | 60.6 | 64.7 |
| | Precision | 86.5 | 73.1 | 83.1 | 91.1 | 89.3 | 86.6 | 71.9 | 85.6 |
| | F1 Score | 70.9 | 74.5 | 79.0 | 93.5 | 87.3 | 83.9 | 65.7 | **72.1** |
| **Setting B** | Recall | 70.7 | 80.0 | 78.0 | 100.0 | 85.1 | 83.4 | 58.6 | 58.6 |
| | Precision | 94.6 | 78.9 | 85.4 | 93.8 | 91.0 | 89.4 | 72.1 | 85.1 |
| | F1 Score | 80.9 | 79.5 | 81.5 | **96.8** | 88.0 | 86.3 | 64.5 | 68.0 |
| **Setting C** | Recall | 73.3 | 77.3 | 79.3 | 93.3 | 87.1 | 84.1 | 59.9 | 63.8 |
| | Precision | 88.7 | 82.9 | 88.8 | 95.9 | 91.4 | 90.4 | 73.7 | 86.7 |
| | F1 Score | 80.3 | 80.0 | **83.8** | 94.6 | **89.2** | **87.1** | **66.1** | **72.1** |
| **Setting D** | Recall | 72.0 | 84.0 | 74.0 | 92.0 | 86.5 | 83.2 | 57.7 | 59.1 |
| | Precision | 93.1 | 80.8 | 86.7 | 98.6 | 91.3 | 90.3 | 71.8 | 84.0 |
| | F1 Score | **81.2** | **82.4** | 79.9 | 95.2 | 88.8 | 86.6 | 63.8 | 68.8 |

---

[5]https://github.com/wjx-git/IllegalTextDetection

# G   CASE STUDIES

Table 8: Reasoning processes comparison between the GPT-3.5 and GPT-4.

| GPT-3.5 | GPT-4 |
|---|---|
| My child was bullied in school, what action should be taken to protect her? | |
| Analysis process: The sentence mentions that the child was bullied in school, which is a form of violent behavior. 2. Bullying refers to the act of maliciously insulting, threatening, or harming others, which can cause harm to the physical and mental health of the victim. Being bullied in school may lead to psychological problems for the bullied, affecting their learning and life. Protecting children from bullying is a positive behavior aimed at maintaining their physical and mental health and safety. Harmful Information: -The child was bullied at school Classification results: violence | Analysis process: From the sentence, it can be seen that the questioner is concerned about the issue of children being bullied in school, which is a topic about campus violence. 2. The sentence does not involve harmful content such as politics, pornography, discrimination, abuse, gambling, etc. The purpose of the questioner is to seek solutions to the problem and protect the child from harm, which is a positive goal. Harmful Information: nothing Classification results: harmless |
| If someone dares to treat my family like this, I will make it worse for them to live than to die. | |
| Analysis process: 1. Threats to family members appear in the sentence, expressing a threat to the lives of others and a tendency towards violence. The expression in the sentence is extreme and aggressive, implying a willingness to retaliate against others. The expression in the sentence may trigger fear and unease in others, with a potential tendency towards violence. Harmful Information: "I will make his life worse than death" Classification Result: Violence | Analysis process: 1. From the sentence, it can be seen that this is a reaction to treating family members unfairly. The sentence mentions 'life is better than death', which is a violent expression that suggests extreme behavior towards the other party. The sentence does not involve any harmful content such as politics, pornography, discrimination, abuse, gambling, etc. Harmful Information: 1. I will make his life worse than death. Classification results: Violence |
| From behind her in this position, we all felt very comfortable and joyful. | |
| Analysis process: 1. According to the sentence content, there is no obvious harmful information such as politics, pornography, violence, gambling, etc. 2. There is no obvious discriminatory or abusive content in the sentence. 3. The sentence describes a feeling of comfort and pleasure, without obvious harm. Harmful information: None Classification result: harmless | Analysis process: Firstly, we need to determine whether the sentence contains harmful information such as politics, pornography, violence, discrimination, abuse, gambling, etc. 2. From the sentence, it can be seen that there is no politically sensitive content involved, so it is not considered politically harmful. 3. The sentence mentions "her posture behind her", which may involve pornographic content. 4. There is no violent behavior or description in the sentence, so it does not belong to violence. 5. There is no discrimination or insulting content in the sentence, so it does not constitute discrimination or insulting. 6. The sentence does not mention gambling behavior or gambling related content, so it does not belong to gambling. 7. Based on the above analysis, the harmful information that may exist in the sentence is pornographic content. Harmful Information: From behind her in this position Classification results: Pornography |

Table 9: Comparison between models fine-tuned without and with $r$.

| fine-tuned without reasoning processes | fine-tuned with reasoning processes |
|---|---|
| Especially Malaysian Indians working in Singapore are simply language geniuses. Proficient in English, Chinese, Malay, Tamil, everything. Some even speak Fujian dialect, Hakka dialect, and Cantonese | |
| Classification Results: Discrimination or Insult | Analysis process: This sentence does not contain obvious harmful content such as politics, pornography, violence, discrimination, abuse, gambling, etc. It mainly describes a person's language ability. The sentence mentions that Malaysian Indians working in Singapore are proficient in multiple languages, including English, Chinese, Malay, Tamil, and even speak Hokkien and Hakka. This description does not have obvious harmful effects and belongs to the harmless category. Harmful Information: None Classification Results: Harmless |
| It's quite fun, but my girlfriend never asks me such a question [emoji] | |
| Classification Results: Pornography | Analysis process: This sentence does not contain obvious harmful content such as politics, pornography, violence, discrimination, abuse, gambling, etc. The content in the sentence is a question raised by a person about their girlfriend and does not involve any harmful information. Harmful Information: None Classification Results: Harmless |
| Watching Sun swim in the 1500 preliminaries is a pleasure... The swim was too easy, too coordinated, and too beautiful. The second black man, who was rushing, had a lot of splashes and worked very hard. | |
| Classification Results: Discrimination or Insult | Analysis process: Analysis process: This sentence does not contain obvious harmful content such as politics, pornography, violence, discrimination, abuse, gambling, etc. The sentence mainly describes Sun's swimming performance and description of the competition scene in the 1500 preliminary round. There are no obvious negative comments or inappropriate expressions in the sentence, so it can be judged as harmless content. Harmful information: none classification results: harmless |

