# OpenReview forum: "Adapting Large Language Models for Content Moderation: Pitfalls in Data Engineering and Supervised Fine-tuning"
_ICLR.cc/2024/Conference — ICLR 2024 Conference Withdrawn Submission_

### Official Review · Reviewer_8Sfa · 2023-11-01

**Soundness:** 3 good
**Presentation:** 2 fair
**Contribution:** 3 good
**Rating:** 6
**Confidence:** 3

**Summary:**

the paper discussed fine tuning llm with a domain specific task, content moderation, as an example. The authors of the paper provide a way to fine tuning with the reasoning process, and show with extensive experiment that it helps pre trained llm models to significantly improve its performance in the content moderation tasks. Overall the paper is clear and easy to follow, and technically sound. See comments below

**Strengths:**

clear implementation details - the first point I like the paper most is it provide details to the extent that it is easy for others to reimplement their method. The prompts, the process, and fine tuning methods, make it a hands on guide to fine tuning domain specific task using llm.

improvement over baseline model - from the experiment I find the purposed reasoning prompt method has significantly improved the two baseline llm's performance. this is useful in that not all scenarios is suitable for cloud llm providers, and it is reasonable self trained or open source models could perform bad on a domain specific task. in general I think the authors of the papers are resolving a real world issue

miscs - ablation studies, discussions, etc are meaningful

**Weaknesses:**

1. from the comparison the purposed methods do not seem to beat gpt 3.5 model by a large margin, and in OOD test CInsult dataset seems the improvement is trivial. although this does not negates the usefulness of the reasoning, it seems to me cloud llm provider still has advantage in most of the tasks

2. some of the tables/figures are not annotated/referenced in the paper like table 2, which is kind of confusing at first glimpse (the first row of table 2 is the setting A in table 1)

3. table 3 is kind of confusing - what do you want to express with the figure? training set performance is similar for reasoning and expansion? how is it related to the main claim of the paper?

**Questions:**

see weakness above

---

### Official Review · Reviewer_65ce · 2023-11-01

**Soundness:** 2 fair
**Presentation:** 2 fair
**Contribution:** 2 fair
**Rating:** 3
**Confidence:** 3

**Summary:**

This paper explores the suitability of using LLMs as models for content moderation. It specifically looks at methods to augment training datasets with reasoning explanations for input-label pairs, which are then fed to an LLM as part of the fine-tuning process. The authors experiment with open-source models ChatGLM2-6B and Baichuan-13B-Chat and compare them to GPT-3.5 as a baseline. Dataset augmentations are conducted using GPT-3.5 and GPT-4 (the LLMs are asked to generate predictions and explanations for input sequences). Using various fine-tuning techniques (using P-tuning and LoRA), the authors show that open-source models fine-tuned on the augmented datasets are competitive with GPT-3.5 on some in-distribution and out-of-distribution categories.

**Strengths:**

The paper discusses the suitability of LLMs to serve as content moderation tools, specifically using open-source models. This is an important area of research since it provides insights into how open-source models can be used in that context and how they compare to commercial models (and hence has potential widespread applicability).

**Weaknesses:**

* The paper claims that their approach results in performances which are "not weaker than" those of GPT-3.5. However, the results in Table 1 do not necessarily support that claim, since performance results are still lower than GPT-3.5 across most of the investigated categories.
* No details on datasets provided: the paper uses a privately collected and annotated dataset for experimentation (in-distribution), however no details on the number of examples and label distributions are provided.
* Related to the above, it would also be helpful to show how many examples have been incorrectly labeled by GPT-3.5 in Setting B and Setting C. How reliably can GPT-3.5 be used in that context?
* The explanations generated by GPT-3.5 to enhance the training datasets have not been manually verified. While I acknowledge that an increase in performance hints at their usefulness, it would be interesting to see how such reasoning samples look like, and whether they truly represent what they are intended to. Furthermore, which prompt formulations were used to obtain the reasoning and label? Did you ask the model to first label and then generate reasoning, or vice versa? Analyzing such details can have a notable impact on performance and should be discussed in more detail.
* Table 2 in the paper is not referred to in the manuscript?
* In Table 3, there are no details on which metrics are being reported, making it difficult to understand the shown results.
* Re. Table 4: The paper assumes that length serves as an indicator for detailedness, which would need to be verified with manual annotations.

**Questions:**

* What is the deviation between ground truth labels and LLM labels for the reasoning creation process?
* The data deduplication step seems confusing: which categories do you use for the clustering? How many cluster labels do you end up with?
* p.4 “Second, the judgment criteria of LLMs are unified, ensuring consistency in reasoning processes across different samples and maintaining the standardization of these processes.” – Can you elaborate on what you mean by that?

---

### Official Review · Reviewer_7bgL · 2023-11-10

**Soundness:** 2 fair
**Presentation:** 2 fair
**Contribution:** 2 fair
**Rating:** 3
**Confidence:** 4

**Summary:**

This work proposes to study the potential of language models in content moderation by fine-tuning them with LLM-generated explanations. Experiments demonstrate that such fine-tuning does indeed help in outperforming prompting alone, while several design choices significantly impact the effectiveness of this fine-tuning pipeline.

**Strengths:**

+ content moderation is an important research topic
+ exploiting large language models for content moderation is a promising direction

**Weaknesses:**

- I wonder if the authors could better clarify which part of the methodology corresponds to addressing the three challenges proposed in the introduction. In addition, how does the proposed approach alleviate the need for quality annotated data?

- What is the unique technical contribution of the proposed pipeline? Data deduplication is not new, and fine-tuning with chain-or-thought paths neither. I'm also not exactly sure if the "weak supervision" part claimed in the methodology is very accurate. It would be great to better highlight in the methodology the parts taken from existing works and the parts uniquely proposed by this work.

- How is self-recheck actually performed in step 2.b? Is this a prompting-based approach? Please provide more details as there is much unclear about this important step.

- I'm not sure about the contributions of Table 1 and Section 3.2. Yes, fine-tuning a smaller model for a specific task outperforms ChatGPT, which might be considered already established. I wonder if the authors could provide more analysis/insights as to this gigantic table.

- One of the major concerns is that there isn't any supervised baseline in this work. While the authors proposed an approach based on supervised fine-tuning and claims for performance gains, it would be better to support it with supervised baselines proposed in recent literature.

- Maybe this is a minor point, but the taxonomy of "content moderation" could be better supported by related literature. There is much research and discussion on what constitutes "content that should be moderated" and the ethical considerations associated with it.

- The related work on content moderation is inadequate. It mostly discusses 2023 works with large language models, potentially overlooking a decade of research on related topics such as misinformation, hate speech, Twitter bots, and more. It is suggested to greatly enrich related work discussions and baselines to better position this work in the context of the research landscape.

**Questions:**

please see above